# Seroprevalence of IgG antibodies against SARS-CoV-2 among the general population and healthcare workers in India, June–July 2021: A population-based cross-sectional study

Manoj V. Murhekar[1]*, Tarun Bhatnagar[1], Jeromie Wesley Vivian Thangaraj[1], V. Saravanakumar[1], Muthusamy Santhosh Kumar[1], Sriram Selvaraju[2], Kiran Rade[3], C. P. Girish Kumar[1], R. Sabarinathan[1], Smita Asthana[4], Rakesh Balachandar[5], Sampada Dipak Bangar[6], Avi Kumar Bansal[7], Jyothi Bhat[8], Debjit Chakraborty[9], Vishal Chopra[10], Dasarathi Das[11], Kangjam Rekha Devi[12], Gaurav Raj Dwivedi[13], Agam Jain[14], S. Muhammad Salim Khan[15], M. Sunil Kumar[16], Avula Laxmaiah[17], Major Madhukar[18], Amarendra Mahapatra[11], Talluri Ramesh[19], Chethana Rangaraju[20], Jyotirmayee Turuk[11], Suresh Yadav[21], Balram Bhargava[22], on behalf of the ICMR serosurveillance group[¶]

1 ICMR–National Institute of Epidemiology, Chennai, India, 2 ICMR–National Institute for Research in Tuberculosis, Chennai, India, 3 WHO Country Office for India, New Delhi, India, 4 ICMR–National Institute of Cancer Prevention and Research, Noida, India, 5 ICMR–National Institute of Occupational Health, Ahmedabad, India, 6 ICMR–National AIDS Research Institute, Pune, India, 7 ICMR–National JALMA Institute for Leprosy & Other Mycobacterial Diseases, Agra, India, 8 ICMR–National Institute of Research in Tribal Health, Jabalpur, India, 9 ICMR–National Institute of Cholera and Enteric Diseases, Kolkata, India, 10 State TB Training and Demonstration Centre, Patiala, India, 11 ICMR–Regional Medical Research Centre, Bhubaneswar, Bhubaneswar, India, 12 ICMR–Regional Medical Research Centre, N. E. Region, Dibrugarh, India, 13 ICMR–Regional Medical Research Centre, Gorakhpur, Gorakhpur, India, 14 State TB Office, Dehradun, India, 15 Government Medical College, Srinagar, Srinagar, India, 16 State TB Training and Demonstration Centre, Thiruvananthapuram, India, 17 ICMR–National Institute of Nutrition, Hyderabad, India, 18 ICMR–Rajendra Memorial Research Institute of Medical Sciences, Patna, India, 19 State TB Office, Hyderabad, India, 20 National Tuberculosis Institute, Bangalore, India, 21 ICMR–National Institute for Implementation Research on Non-Communicable Diseases, Jodhpur, India, 22 Indian Council of Medical Research, New Delhi, India

☯ These authors contributed equally to this work.
¶ Membership of the ICMR serosurveillance group is provided in the Acknowledgements.
* mmurhekar@nieicmr.org.in

**Data Availability Statement:** The authors confirm that, for IHEC-approved reasons, some access

## Abstract

### Background

India began COVID-19 vaccination in January 2021, initially targeting healthcare and front-line workers. The vaccination strategy was expanded in a phased manner and currently covers all individuals aged 18 years and above. India experienced a severe second wave of COVID-19 during March–June 2021. We conducted a fourth nationwide serosurvey to estimate prevalence of SARS-CoV-2 antibodies in the general population aged ≥6 years and healthcare workers (HCWs).

restrictions apply to the data underlying the findings. An individual may email to ICMR Data Repository office (harpreets.hq@icmr.gov.in) to request the data. Given the nature of these data, potential users will be asked to sign a data sharing agreement. This is not intended to restrict access, but to ensure that requests are for ethical research purposes and that any analyses undertaken will not compromise the confidentiality of individual participants, and are not for commercial purposes.

**Funding:** MVM received the funding from Indian Council of Medical Research, New Delhi. The funders were involved in study design, and the decision to publish and preparation of the manuscript.

**Competing interests:** The authors have declared that no competing interests exist.

**Abbreviations:** N, nucleocapsid protein; HCW, healthcare worker.

## Methods and findings

We did a cross-sectional study between 14 June and 6 July 2021 in the same 70 districts across 20 states and 1 union territory where 3 previous rounds of serosurveys were conducted. From each district, 10 clusters (villages in rural areas and wards in urban areas) were selected by the probability proportional to population size method. From each district, a minimum of 400 individuals aged ≥6 years from the general population (40 individuals from each cluster) and 100 HCWs from the district public health facilities were included. The serum samples were tested for the presence of IgG antibodies against S1-RBD and nucleocapsid protein of SARS-CoV-2 using chemiluminescence immunoassay. We estimated the weighted and test-adjusted seroprevalence of IgG antibodies against SARS-CoV-2, along with 95% CIs, based on the presence of antibodies to S1-RBD and/or nucleocapsid protein. Of the 28,975 individuals who participated in the survey, 2,892 (10%) were aged 6–9 years, 5,798 (20%) were aged 10–17 years, and 20,285 (70%) were aged ≥18 years; 15,160 (52.3%) participants were female, and 21,794 (75.2%) resided in rural areas. The weighted and test-adjusted prevalence of IgG antibodies against S1-RBD and/or nucleocapsid protein among the general population aged ≥6 years was 67.6% (95% CI 66.4% to 68.7%). Seroprevalence increased with age ($p < 0.001$) and was not different in rural and urban areas ($p = 0.822$). Compared to unvaccinated adults (62.3%, 95% CI 60.9% to 63.7%), seroprevalence was significantly higher among individuals who had received 1 vaccine dose (81.0%, 95% CI 79.6% to 82.3%, $p < 0.001$) and 2 vaccine doses (89.8%, 95% CI 88.4% to 91.1%, $p < 0.001$). The seroprevalence of IgG antibodies among 7,252 HCWs was 85.2% (95% CI 83.5% to 86.7%). Important limitations of the study include the survey design, which was aimed to estimate seroprevalence at the national level and not at a sub-national level, and the non-participation of 19% of eligible individuals in the survey.

## Conclusions

Nearly two-thirds of individuals aged ≥6 years from the general population and 85% of HCWs had antibodies against SARS-CoV-2 by June–July 2021 in India. As one-third of the population is still seronegative, it is necessary to accelerate the coverage of COVID-19 vaccination among adults and continue adherence to non-pharmaceutical interventions.

### Author summary

#### Why was this study done?

- Earlier nationwide COVID-19 serosurveys conducted in India indicated an increase in seroprevalence from 0.73% (95% CI 0.34% to 1.13%) in May–June 2020 to 6.6% (95% CI 5.8% to 7.4%) in September–October 2020 and 24.1% (95% CI 23.0% to 25.3%) in December 2020–January 2021.

- India began COVID-19 vaccination in January 2021, initially targeting healthcare and frontline workers. The vaccination strategy was expanded in a phased manner and currently covers all individuals aged 18 years and above.

- India witnessed a severe second wave of COVID-19 in March–June 2021.

**What did the researchers do and find?**

- The fourth nationwide serosurvey indicated that about two-thirds of India's population aged ≥6 years had antibodies against SARS-CoV-2 by June–July 2021.

- Seroprevalence increased with age, but was not different in urban slum, urban non-slum, and rural areas.

- Seroprevalence was significantly higher among individuals who had received 2 doses of COVID-19 vaccine compared to unvaccinated individuals.

- About 85% of healthcare workers working in district-level health facilities had antibodies against SARS-CoV-2.

**What do these findings mean?**

- The substantial seroprevalence of anti-SARS-CoV-2 antibodies in the Indian population should provide some measure of protection against future waves of COVID-19 in the country.

- About one-third of the population in India did not have detectable antibodies against SARS-CoV-2 by June–July 2021. It is therefore necessary to accelerate the coverage of COVID-19 vaccination among adults.

## Introduction

With more than 30 million cases (21,961 cases per million population) and 0.48 million deaths (289 per million population) as of 5 July 2021, India has the second largest number of COVID-19 cases reported globally [1]. India experienced a severe second wave of COVID-19 in March–June 2021, affecting all states of India [2]. Repeated cross-sectional serosurveys in the same geographical location are useful to monitor the trends of seroprevalence over time and to provide evidence for public health decision-making to plan the response [3]. Serial serosurveys conducted in 70 districts spread across 20 Indian states and 1 union territory (hereafter referred to collectively as states) prior to the introduction of COVID-19 vaccination indicated that the seroprevalence in India increased from 0.73% (95% CI 0.34% to 1.13%) in May–June 2020 to 6.6% (95% CI 5.8% to 7.4%) in September–October 2020 and 24.1% (95% CI 23.0% to 25.3%) in December 2020–January 2021 [4–6]. About 25% of healthcare workers (HCWs) working in sub-district health facilities in these 70 districts had evidence of IgG antibodies against SARS-CoV-2 in December 2020–January 2021 [6]. The previous nationwide serosurveys in the general population were conducted among individuals aged 10 years and above and do not provide information about seroprevalence among children below 10 years of age.

On 16 January 2021, India initiated COVID-19 vaccination with BBV152 (Covaxin; Bharat Biotech International, Hyderabad) and ChAdOx1 nCoV-19 (Covishield, Serum Institute of India, Pune) for healthcare and frontline workers. The vaccination strategy was expanded in a phased manner to cover individuals aged 60 years and above and those between 45 and 59 years with

specified comorbidities (phase 2, 1 March 2021), all individuals aged 45 years and above (phase 3, 1 April 2021), and all individuals aged 18 years and above (phase 4, 1 May 2021) [7].

We conducted the fourth round of national serosurvey to estimate the age-specific prevalence of SARS-CoV-2 antibodies in the general population and among HCWs.

## Methods

This study is reported as per the Strengthening the Reporting of Observational Studies in Epidemiology guidelines (S1 STROBE Checklist). The study had a protocol with analysis plan (S1 Protocol).

### Study design and participants

We conducted a cross-sectional survey between 14 June and 6 July 2021 in the same 70 districts spread across 21 Indian states where 3 previous rounds of serosurveys were conducted [4–6]. We planned to enrol 28,000 participants from the general population with 19,600 individuals aged ≥18 years, 6,800 children aged 10–17 years, and 2,800 children aged 6–9 years (S1 Protocol). From each district, we selected 10 clusters (wards in urban areas and villages in rural areas) using the probability proportional to population size method. The survey teams selected 4 random starting points within each of the selected clusters. Starting from a random starting point, the teams visited consecutive households and listed all household members aged 6 years and above who were permanent residents of the area. Eligible individuals present in the household were invited to participate in the survey. From each random location, at least 10 consenting individuals (1 aged 6–9 years, 2 aged 10–17 years, and 7 aged ≥18 years) were enrolled in the survey. Enrolment of a minimum number of individuals in each age group ensured that the overall distribution of the sampled population was comparable to the age structure of the population in India [8]. Thus, a minimum of 40 individuals from each cluster and 400 individuals from each district were enrolled.

We planned to enrol 7,000 HCWs (S1 Protocol). We enrolled at least 100 HCWs from each of the 70 districts selected for the general population survey. All of the HCWs (doctors, nurses, para medical staff, and lab staff) working in the district headquarters hospital of the selected study district and present on the day of the visit of the survey team were invited to participate in the survey, and consenting HCWs were enrolled in the study. If the sample size of 100 could not be achieved in the district headquarters hospital, the team selected the nearest sub-district-level public health facility to enrol additional HCWs.

### Procedures

We interviewed eligible consenting participants to collect information about demographic details, history of symptoms suggestive of COVID-19 (e.g., fever, cough, shortness of breath, sore throat, new loss of taste or smell, fatigue) since 1 January 2021, COVID-19 testing, and COVID-19 vaccination. Three millilitres of venous blood was collected from each participant, and serum samples were transported to the ICMR–National Institute of Epidemiology, Chennai, under cold chain.

We tested the serum samples for the presence of IgG antibodies against S1-RBD (ADVIA Centaur XP/XPT, Siemens Healthineers, Munich, Germany) and nucleocapsid protein (Abbott ARCHITECT, Abbott Laboratories, Abbott Park, IL, US) of SARS-CoV-2 using chemiluminescence immunoassay, as per the manufacturers' instructions. The Siemens assay is a quantitative antibody assay with an analytical measuring interval of 0.50–150.0; samples with an index value of ≥1 are considered as reactive. The assay has sensitivity of 96.4% (95% CI 92.7% to 98.5%) 21 days after PCR confirmation of SARS-CoV-2 infection, and specificity

of 99.90% (95% CI 99.64% to 99.99%) [9]. The Abbott assay for IgG antibodies against nucleo-capsid protein is a qualitative assay and has a sensitivity of 100.0% and specificity of 99.6% 14 days after PCR confirmation [10]. As a part of quality control, 10% of positive serum samples and an equal number of negative serum samples were re-tested using the same assay.

We also separately estimated the performance of the 2 assays by testing 100 pre-pandemic sera samples collected as a part of acute fever surveillance during 2016 and 140 samples from patients with laboratory-confirmed COVID-19 collected 30–240 days after PCR confirmation [11]. We estimated a specificity of 99.0% (95% CI 94.6% to 100.0%) and sensitivity of 80.0% (95% CI 72.4% to 86.3%) for the S1-RBD assay and a specificity of 100% (95% CI 96.4% to 100%) and sensitivity of 61.4% (95% CI 52.8% to 69.5%) for the nucleocapsid protein assay in detecting historical infection.

## Data analysis

The characteristics of study participants were described as proportions. Individuals whose serum sample was positive for IgG antibodies against S1-RBD and/or nucleocapsid protein were considered as seropositive. We calculated design weights as the product of the inverse of the sampling fraction for the selection of districts and the selection of clusters within each district. We estimated the weighted seroprevalence of IgG antibodies, along with 95% confidence intervals (CIs), using a random-effects model to account for cluster sampling (S3 Text). The weighted seroprevalence was further adjusted for the joint sensitivity and specificity of the 2 assays using the sensitivities and specificities estimated by the manufacturer [12]. In addition to the analysis prespecified in the protocol, we conducted a sensitivity analysis estimating seroprevalence using the lowest sensitivity and specificity of the 2 assays estimated through the external validation studies, as well as considering the sensitivity and specificity estimated during in-house validation (S1 Text). We also estimated seroprevalence by selected demographic and COVID-19-related characteristics of the study participants.

We estimated the total number of individuals infected with SARS-CoV-2 at the national level. To estimate the total number of children infected with SARS-CoV-2, we applied the weighted seroprevalence of IgG antibodies against SARS-CoV-2 among children aged 6–17 years to the total population of children aged 6–17 years. To estimate the total number of infections among individuals aged ≥18 years, we considered 2 scenarios. First, we applied the weighted seroprevalence of IgG antibodies against SARS-CoV-2 among unvaccinated individuals aged ≥18 years to the total population of unvaccinated individuals aged ≥18 years. Second, we applied the weighted seroprevalence of IgG antibodies against SARS-CoV-2 among unvaccinated individuals to the total population aged ≥18 years. The infection-to-case ratio (ICR) was calculated by dividing the estimated number of SARS-CoV-2 infections by the cumulative number of COVID-19 cases reported in India 1 and 2 weeks before the median survey date (23 June 2020), assuming IgG antibodies start appearing between 5 and 15 days post-infection [13].

## Protection of human participants

Written informed consent was obtained from individuals aged ≥18 years. For children aged between 7 and 17 years, we obtained assent from the children and written consent from their parents or guardians, while only parental consent was obtained for children aged 6 years. The Institutional Human Ethics Committee of the ICMR–National Institute of Epidemiology, Chennai, approved the study protocol.

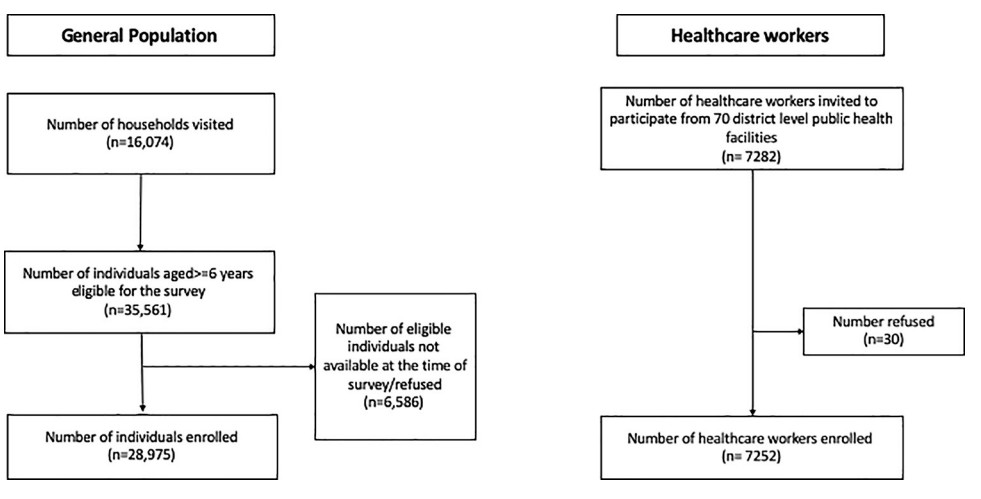

**Fig 1. Flowchart of participant enrolment.**

## Results

### Seroprevalence among the general population

The survey teams visited 16,074 households from 700 clusters in 70 Indian districts. Of the 35,561 individuals aged ≥6 years residing in these households, 28,975 (81.5%) consented to participated in the survey (Fig 1).

Of the 28,975 individuals who participated in the survey, 2,892 (10%) were aged 6–9 years, 5,798 (20%) were aged 10–17 years, and 20,285 (70%) were aged ≥18 years; 15,160 (52.3%) participants were female, and 21,794 (75.2%) resided in rural areas (Table 1). In total, 4,372 (15.1%) of the 28,956 individuals with data on COVID-19 testing reported being previously tested for COVID-19, of whom 782 (17.9%) reported a positive test result. Of the 20,268 adult participants with data on vaccination, 5,038 (24.8%) and 2,631 (13.0%) reported receipt of 1 and 2 doses of COVID-19 vaccines, respectively, while the remaining 12,599 (62.2%) were unvaccinated. Most vaccinated individuals ($n$ = 6,945, 90.6%) had received the Covishield vaccine (Table 1). The median interval between the receipt of the first dose and the date of sample collection was 36 days (IQR 14–68), and between receipt of the second dose and sample collection was 60 days (IQR 34–98).

Of the 28,975 sera tested, 11,289 (38.9%) had IgG antibodies against nucleocapsid protein, 18,388 (63.5%) had antibodies against S1-RBD, and 19,336 (66.7%) had antibodies against nucleocapsid protein and/or S1-RBD. The weighted prevalence of IgG antibodies against S1-RBD and/or nucleocapsid protein was 67.6% (95% CI 66.4% to 68.7%) after adjusting for assay characteristics (Table 2). The weighted prevalence of IgG antibodies against nucleocapsid protein was 38.3% (95% CI 37.0% to 39.5%) (Table 2). Anti-nucleocapsid-protein (anti-N) seropositivity was not different by age, gender, or area of residence (S1 Table). Also, around one-third of the individuals vaccinated with Covishield had anti-N IgG antibodies.

The overall seropositivity in states ranged between 44.3% (Kerala) and 80.0% (Madhya Pradesh) (S2 Table). The seropositivity rate was heterogenous among the 70 districts, ranging between 40.5% (Ernakulam, Kerala) and 86.8% (Buxar, Bihar) (S3 Table). Compared to December 2020–January 2021, the seropositivity rate among unvaccinated individuals aged 10 years and above in June–July 2021 had increased by 1.2- to 1.9-fold in 21 (30.0%) districts, 2- to 3-fold in 31 (44.3%) districts, and >3-fold in 18 (25.7%) districts (S4 Table).

Seroprevalence showed a rising trend with age ($p < 0.001$). Among children aged 6–9 years and 10–17 years, respectively 57.2% (95% CI 55.0% to 59.4%) and 61.6% (95% CI 59.8% to

**Table 1. Participant characteristics.**

| Characteristic | General population, *n* (%) | Healthcare workers, *n* (%) |
|---|---|---|
| **Age** | ***n* = 28,975** | ***n* = 7,252** |
| 6–9 years | 2,892 (10.0) | — |
| 10–17 years | 5,798 (20.0) | — |
| 18–44 years | 12,522 (43.2) | 5,133 (70.8) |
| 45–60 years | 5,545 (19.1) | 1,997 (27.5) |
| >60 years | 2,218 (7.7) | 122 (1.7) |
| **Gender** | ***n* = 28,975** | ***n* = 7,252** |
| Male | 13,783 (47.6) | 3,523 (48.6) |
| Female | 15,160 (52.3) | 3,722 (51.3) |
| Other | 32 (0.1) | 7 (0.1) |
| **Area of residence** | ***n* = 28,975** | — |
| Rural | 21,794 (75.2) | — |
| Urban non-slum | 5,266 (18.2) | — |
| Urban slum | 1,915 (6.6) | — |
| **COVID-19-related symptoms** | ***n* = 28,975** | ***n* = 7,252** |
| History of COVID-19 symptoms since January 2021 | 1,748 (6.0) | 925 (12.8) |
| **COVID-19-related treatment** | ***n* = 1,729** | ***n* = 925** |
| Medical care sought for symptomatic cases | 574 (33.2) | 558 (60.3) |
| History of hospitalisation | 140 (24.4) | 171 (30.6) |
| **COVID-19 contact** | ***n* = 28,956** | ***n* = 7,252** |
| History of contact with a known COVID-19 case | 2,129 (7.3) | 4,562 (62.9) |
| **COVID-19 testing** | ***n* = 28,956** | ***n* = 7,252** |
| Previously tested for COVID-19 | 4,372 (15.1) | 4,892 (67.4) |
| Previous positive COVID-19 test | 782 (17.9) | 1,354 (27.7) |
| **COVID-19 vaccination status among adults** | ***n* = 20,268** | ***n* = 7,252** |
| 0 dose | 12,599 (62.2) | 759 (10.5) |
| 1 dose | 5,038 (24.8) | 972 (13.4) |
| 2 doses | 2,631 (13.0) | 5,521 (76.1) |
| **Type of vaccine** | ***n* = 7,669** | ***n* = 6,493** |
| Covaxin | 587 (7.7) | 498 (7.7) |
| Covishield | 6,945 (90.6) | 5,973 (92.0) |
| Sputnik | 18 (0.2) | 6 (0.1) |
| Don't know | 119 (1.6) | 16 (0.2) |

63.3%) had antibodies against SARS-CoV-2. The seroprevalence among adults was 66.7% (95% CI 65.3% to 68.0%) for individuals aged 18–44 years and 77.6% (95% CI 76.1% to 79.0%) for individuals aged 45–60 years. Seroprevalence was not different in rural, urban non-slum, and urban slum areas ($p = 0.822$). Among the 487 unvaccinated individuals with a history of laboratory-confirmed COVID-19 infection, 88.0% (95% CI 83.0% to 91.8%) had detectable antibodies against SARS-CoV-2 (Table 3).

Compared to the seroprevalence in unvaccinated adults, seroprevalence was significantly higher among individuals who received 1 dose (81.0%, 95% CI 79.6% to 82.3%, $p = 0.001$) or 2 doses (89.8%, 95% CI 88.4% to 91.1%, $p = 0.001$) of COVID-19 vaccine. This difference was observed across all age groups, both genders, and all areas of residence among the individuals in the general population (Table 3). Seroprevalence was higher among individuals who reported receipt of Covishield (85.2%, 95% CI 83.8% to 86.5%) than among those who reported receipt of Covaxin (80.2%, 95% CI 76.1% to 83.8%) ($p = 0.004$) (Table 3). After 1

**Table 2. Seroprevalence (percent) of IgG antibodies against SARS-CoV-2, India, June–July 2021.**

| Measure | General population aged ≥6 years | | | Healthcare workers | | |
|---|---|---|---|---|---|---|
| | Anti-N antibodies | Anti-S1-RBD antibodies | Anti-N and/or anti-S-RBD antibodies | Anti-N antibodies | Anti-S1-RBD antibodies | Anti-N and/or anti-S-RBD antibodies |
| Number of individuals tested | 28,975 | 28,975 | 28,975 | 7,252 | 7,252 | 7,252 |
| Number positive | 11,289 | 18,388 | 19,336 | 2,305 | 6,112 | 6,186 |
| Unweighted prevalence*, percent (95% CI) | 38.9 (37.9–40.1) | 63.5 (62.3–64.6) | 66.7 (65.6–67.8) | 31.8 (29.7–34.0) | 84.3 (82.5–85.9) | 85.3 (83.6–86.8) |
| Weighted prevalence**, percent (95% CI) | 38.5 (37.3–39.7) | 64.4 (63.2–65.6) | 66.6 (65.3–67.9) | — | — | — |
| Adjusted prevalence***, percent (95% CI) | 38.3 (37.0–39.5) | 66.8 (65.5–68.0) | 67.6 (66.4–68.7) | 31.5 (29.4–33.7) | 87.4 (85.6–89.1) | 85.2 (83.5–86.7) |

*Adjusted for clustering.

**Weighted for design weights.

***Adjusted for test performance.

N, nucleocapsid protein.

dose, the seroprevalence was 80.8% (95% CI 75.7% to 85.0%) among Covaxin recipients and 82.0% (95% CI 80.3% to 83.6%) among Covishield recipients. Individuals who reported receipt of 2 doses had higher seroprevalence, 86.3% and 90.3% for Covaxin and Covishield recipients, respectively (Table 4).

Using the sensitivities and specificities estimated from the in-house validation and external validation studies, the overall seroprevalence of IgG antibodies against SARS-CoV-2 was 73.2% (95% CI 71.9% to 74.4%) considering the joint sensitivity and specificity estimated from the in-house validation and 68.0% (95% CI 66.8% to 69.1%) considering the joint sensitivity and specificity estimated in external validation studies. We estimated that there were 642,751,546 to 807,395,611 SARS-CoV-2 infections in India by mid-June 2021. With 29,088,245 and 29,632,302 cases reported by 9 June and 16 June 2021, respectively, the number of infections per reported COVID-19 case ranged between 21.7 and 27.8 (S5 Table).

## Seroprevalence among HCWs

We enrolled 7,252 HCWs from the district public hospitals of the 70 districts selected for the general population survey (Fig 1). Most HCWs (n = 5,133, 70.8%) were aged 18–44 years, and 51.3% (n = 3,722) were female. Of the 4,892 (67.4%) HCWs who reported a history of COVID-19 testing, 1,354 (27.7%) had had a positive test result. Overall, 89.5% (n = 6,493) reported a history of COVID-19 vaccination, while the remaining individuals were unvaccinated (Table 1).

Of the 7,252 HCWs, 6,186 (85.3%) had antibodies against nucleocapsid protein and/or S1-RBD, with a seroprevalence of 85.2% (95% CI 83.5% to 86.7%) after adjusting for assay characteristics (Table 2). Seroprevalence did not differ by age group or gender (Table 3). Seroprevalence was higher among HCWs who reported receipt of 1 dose (87.7%, 95% CI 85.0% to 89.9%, p < 0.001) or 2 doses (88.6%, 95% CI 87.1% to 90.1%, p < 0.001) of COVID-19 vaccine as compared to unvaccinated individuals (64.8%, 95% CI 60.1% to 69.2%) (Table 3). After one dose the seroprevalence was 79.0% (95% CI 69.5% to 86.0%) among Covaxin recipients and 88.5% (95% CI 85.5% to 91.1%) among Covishield recipients (Table 5).

## Discussion

The results from the fourth nationwide serosurvey indicate that about two-thirds of India's population aged 6 years and above had antibodies against SARS-CoV-2 by June–July 2021.

**Table 3. Seroprevalence of IgG antibodies against SARS-CoV-2 by selected characteristics, June–July 2021.**

| Characteristic | General population | | | Healthcare workers | | |
|---|---|---|---|---|---|---|
| | Number tested | Number positive (anti-N and/or anti-S1-RBD antibodies) | Weighted and test-performance-adjusted seroprevalence, percent (95% CI) | Number tested | Number positive (anti-N and/or anti-S1-RBD antibodies) | Test-performance-adjusted seroprevalence, percent (95% CI) |
| **Age** | | | | | | |
| 6–9 years | 2,892 | 1,635 | 57.2 (55.0–59.4) | — | — | — |
| 10–17 years | 5,798 | 3,584 | 61.6 (59.8–63.3) | — | — | — |
| 18–44 years | 12,522 | 8,245 | 66.7 (65.3–68.0) | 5,133 | 4,401 | 86.5 (84.9–88.0) |
| 45–60 years | 5,545 | 4,217 | 77.6 (76.1–79.0) | 1,997 | 1,686 | 85.1 (83.0–87.1) |
| >60 years | 2,218 | 1,655 | 76.7 (74.6–78.7) | 122 | 99 | 80.4 (71.9–86.8) |
| **Gender** | | | | | | |
| Male | 13,783 | 9,018 | 65.8 (64.4–67.1) | 3,523 | 3,024 | 86.2 (84.4–87.8) |
| Female | 15,160 | 10,295 | 69.2 (67.9–70.5) | 3,722 | 3,157 | 85.9 (84.1–87.6) |
| Other | 32 | 23 | 83.4 (59.1–94.6) | 7 | 5 | 66.4 (26.9–91.3) |
| **Area of residence** | | | | | | |
| Rural | 21,794 | 14,398 | 66.7 (65.4–68.1) | — | — | — |
| Urban non-slum | 5,266 | 3,587 | 69.1 (66.6–71.6) | — | — | — |
| Urban slum | 1,915 | 1,351 | 71.0 (66.8–74.7) | — | — | — |
| **History of COVID-19-related symptoms since 1 January 2021** | | | | | | |
| Yes | 1,748 | 1,262 | 76.8 (74.4–79.0) | 925 | 838 | 85.2 (83.6–86.8) |
| No | 27,227 | 18,074 | 66.9 (65.7–68.1) | 6,327 | 5,348 | 91.5 (89.2–93.2) |
| **Previously tested for COVID-19** | | | | | | |
| Yes | 4,372 | 3,196 | 78.7 (77.1–80.2) | 4,892 | 4,194 | 86.9 (85.2–88.4) |
| No | 24,584 | 16,127 | 65.6 (64.4–66.9) | 2,360 | 1,992 | 84.2 (81.9–86.3) |
| **Previous COVID-19 test result** | | | | | | |
| Reported positive for COVID-19 | 782 | 674 | 88.9 (86.8–90.8) | 1,354 | 1,275 | 94.8 (93.4–96.0) |
| Reported negative for COVID-19 | 3,419 | 2,425 | 75.2 (73.2–77.0) | 3,395 | 2,789 | 83.3 (81.1–85.3) |
| Don't know | 171 | 97 | 71.1 (59.2–80.5) | 143 | 130 | 90.1 (80.6–95.2) |
| **COVID-19 vaccination status among adults** | | | | | | |
| 0 dose | 12,599 | 7,758 | 62.3 (60.9–63.7) | 759 | 507 | 64.8 (60.1–69.2) |
| 1 dose | 5,038 | 4,016 | 81.0 (79.6–82.3) | 972 | 834 | 87.7 (85.0–89.9) |
| 2 doses | 2,631 | 2,331 | 89.8 (88.4–91.1) | 5,521 | 4,845 | 88.6 (87.1–90.1) |
| **Timing of blood sample collection** | | | | | | |
| Less than 21 days after first dose | 1,711 | 1,242 | 73.5 (70.6–76.2) | 191 | 151 | 78.0 (70.7–83.9) |
| 21 days or more after first dose | 3,327 | 2,774 | 85.9 (84.3–87.4) | 781 | 683 | 89.8 (87.2–92.1) |
| 7 days or more after second dose | 2,630 | 2,330 | 90.4 (88.9–91.7) | 5,513 | 4,837 | 88.6 (87.1–90.1) |
| **Vaccine type** | | | | | | |
| Covaxin | 587 | 473 | 80.2 (76.1–83.8) | 498 | 428 | 86.5 (82.7–89.5) |
| Covishield | 6,945 | 5,751 | 85.2 (83.8–86.5) | 5,973 | 5,229 | 88.6 (87.0–90.1) |
| **Previously positive for COVID-19, by vaccination status** | | | | | | |
| 0 dose | 487 | 402 | 88.0 (83.0–91.8) | 140 | 116 | 83.6 (76.0–89.2) |
| 1 dose | 145 | 134 | 95.0 (90.6–97.4) | 154 | 146 | 95.2 (90.4–97.7) |
| 2 doses | 150 | 138 | 94.0 (88.2–97.1) | 1,060 | 1,013 | 96.1 (94.4–97.4) |

N, nucleocapsid protein.

**Table 4. Seroprevalence of IgG antibodies against SARS-CoV-2 in the general population by vaccination status, June–July 2021.**

| Characteristic | Unvaccinated | | | Vaccinated with 1 dose | | | Vaccinated with 2 doses | | |
|---|---|---|---|---|---|---|---|---|---|
| | Number tested | Number positive (anti-N and/or anti-S1-RBD antibodies) | Cluster- and test-adjusted seroprevalence, percent (95% CI) | Number tested | Number positive (anti-N and/or anti-S1-RBD antibodies) | Cluster- and test-adjusted seroprevalence, percent (95% CI) | Number tested | Number positive (anti-N and/or anti-S1-RBD antibodies) | Cluster- and test-adjusted seroprevalence, percent (95% CI) |
| **Age** | | | | | | | | | |
| 18–44 years | 8,986 | 5,381 | 60.8 (59.2–62.3) | 2,426 | 1,896 | 80.4 (78.2–82.4) | 1,096 | 958 | 89.1 (86.7–91.3) |
| 45–60 years | 2,701 | 1,778 | 67.0 (64.8–69.1) | 1,830 | 1,521 | 85.5 (83.4–87.3) | 1,013 | 917 | 91.9 (89.7–93.6) |
| >60 years | 912 | 599 | 67.0 (63.6–70.4) | 782 | 599 | 78.9 (75.4–82.0) | 522 | 456 | 88.7 (85.4–91.5) |
| **Gender** | | | | | | | | | |
| Male | 10,040 | 5,962 | 59.6 (57.8–61.4) | 2,598 | 2,050 | 81.0 (79.0–82.9) | 1,136 | 999 | 89.4 (87.1–91.5) |
| Female | 11,223 | 6,998 | 64.9 (63.2–66.4) | 2,436 | 1,962 | 83.1 (81.1–84.9) | 1,493 | 1,330 | 90.7 (88.6–92.3) |
| Other | 26 | 17 | 74.0 (44.9–90.8) | 4 | 4 | — | 2 | 2 | — |
| **Area of residence** | | | | | | | | | |
| Rural | 3,702 | 2,280 | 62.1 (60.5–63.7) | 1,046 | 846 | 81.1 (79.2–82.9) | 515 | 459 | 90.1 (88.0–91.7) |
| Urban non-slum | 16,236 | 9,832 | 63.6 (60.2–66.8) | 3,603 | 2,840 | 83.2 (79.8–86.1) | 1,941 | 1,716 | 90.4 (86.9–93.0) |
| Urban slum | 1,351 | 865 | 64.6 (59.0–69.8) | 389 | 330 | 87.1 (82.1–90.9) | 175 | 156 | 90.9 (84.7–94.7) |
| **Previous COVID-19 test result** | | | | | | | | | |
| Reported positive for COVID-19 | 487 | 402 | 86.0 (82.2–89.1) | 145 | 134 | 94.7 (90.1–97.2) | 150 | 138 | 93.8 (88.6–96.7) |
| Reported negative for COVID-19 | 1,732 | 1,051 | 62.9 (59.7–66.1) | 962 | 739 | 80.1 (76.9–83.0) | 724 | 634 | 89.4 (86.5–91.8) |
| Don't know | 119 | 58 | 60.2 (46.5–72.5) | 36 | 28 | 84.1 (67.5–93.1) | 16 | 11 | 66.7 (37.4–87.0) |
| **Vaccine type** | | | | | | | | | |
| Covaxin | — | — | — | 385 | 302 | 80.8 (75.7–85.0) | 202 | 171 | 86.3 (80.4–90.8) |
| Covishield | — | — | — | 4,565 | 3,636 | 82.0 (80.3–83.6) | 2,380 | 2,115 | 90.3 (88.5–91.8) |
| **Individuals with optimal interval between vaccination and blood sample collection (21 days or more after first dose, 7 days or more after second dose)** | | | | | | | | | |
| Covaxin | — | — | — | 223 | 186 | 84.3 (78.4–88.9) | 201 | 170 | 86.3 (80.3–90.7) |
| Covishield | — | — | — | 3,045 | 2,535 | 84.8 (83.1–86.5) | 2,380 | 2,115 | 90.3 (88.5–91.8) |

N, nucleocapsid protein.

Seroprevalence increased with age. Seroprevalence was comparable in rural, urban non-slum, and urban slum areas. The majority of HCWs working in district-level health facilities were positive for IgG antibodies. These findings have important implications for the future trajectory of COVID-19 in India.

The overall prevalence of IgG antibodies against SARS-CoV-2 increased from 24.1% in December 2020–January 2021 to 67.6% in June–July 2021 (S6 Table). This increase in seroprevalence could be due to natural infection as well as COVID-19 vaccination. The seroprevalence among unvaccinated adults in June–July 2021 was 62.3%, compared to 24.3% in December 2020–January 2021. This finding indicates that a large proportion of the increase in seroprevalence was due to natural infection during the second wave of COVID-19 in India in March–June 2021. Around 38% of individuals had anti-N antibodies indicating recent transmission of SARS-CoV-2 [14]. During the second wave of COVID-19, more than 20 million COVID-19 cases were reported from India [2], with the delta variant being a predominant circulating variant of concern [15].

**Table 5. Seroprevalence of IgG antibodies against SARS-CoV-2 among healthcare workers by vaccination status, June–July 2021.**

| Characteristic | Unvaccinated | | | Vaccinated with 1 dose | | | Vaccinated with 2 doses | | |
|---|---|---|---|---|---|---|---|---|---|
| | Number tested | Number positive (anti-N and/or anti-S1-RBD antibodies) | Cluster- and test-adjusted seroprevalence, percent (95% CI) | Number tested | Number positive (anti-N and/or anti-S1-RBD antibodies) | Cluster- and test-adjusted seroprevalence, percent (95% CI) | Number tested | Number positive (anti-N and/or anti-S1-RBD antibodies) | Cluster- and test-adjusted seroprevalence, percent (95% CI) |
| **Age** | | | | | | | | | |
| 18–44 years | 651 | 429 | 65.6 (60.5–70.5) | 828 | 712 | 88.2 (84.8–91.0) | 3,654 | 3,260 | 90.1 (88.5–91.5) |
| 45–60 years | 105 | 77 | 73.8 (63.3–82.0) | 142 | 120 | 85.1 (77.7–90.5) | 1,750 | 1,489 | 86.1 (83.7–88.2) |
| >60 years | 3 | 1 | 35.6 (3.8–87.4) | 2 | 2 | — | 117 | 96 | 81.9 (73.2–88.2) |
| **Gender** | | | | | | | | | |
| Male | 343 | 230 | 66.0 (59.7–72.0) | 451 | 392 | 88.3 (84.4–91.5) | 2,729 | 2,402 | 88.7 (86.9–90.4) |
| Female | 414 | 277 | 67.4 (61.4–72.9) | 521 | 442 | 87.1 (83.0–90.4) | 2,787 | 2,438 | 88.5 (86.7–90.3) |
| Other | 2 | 0 | — | — | — | — | 5 | 5 | — |
| **Results of COVID-19 testing** | | | | | | | | | |
| Reported positive for COVID-19 | 140 | 116 | 85.0 (77.0–90.6) | 154 | 146 | 95.6 (90.9–97.9) | 1,060 | 1,013 | 96.3 (94.9–97.3) |
| Reported negative for COVID-19 | 267 | 156 | 58.7 (50.5–66.4) | 491 | 408 | 85.9 (80.9–89.7) | 2,637 | 2,225 | 85.4 (83.1–87.4) |
| Don't know | 16 | 11 | 71.9 (37.0–91.7) | 15 | 15 | — | 112 | 104 | 92.1 (81.9–96.8) |
| **Vaccine type** | | | | | | | | | |
| Covaxin | — | — | — | 127 | 96 | 79.0 (69.5–86.0) | 371 | 332 | 89.0 (84.8–92.2) |
| Covishield | — | — | — | 841 | 734 | 88.5 (85.5–91.1) | 5,132 | 4,495 | 88.6 (87.0–90.1) |
| **Individuals with optimal interval between vaccination and blood sample collection (21 days or more after first dose, 7 days or more after second dose)** | | | | | | | | | |
| Covaxin | — | — | — | 103 | 81 | 83.8 (73.5–90.6) | 368 | 329 | 88.9 (84.7–92.1) |
| Covishield | — | — | — | 675 | 599 | 91.0 (87.4–93.6) | 5,127 | 4,490 | 88.6 (87.0–90.1) |

N, nucleocapsid protein.

The increase in seroprevalence was observed in all age groups, including children aged 10–17 years (2.2-fold) (S6 Table). Children aged 6–9 years were not covered during the previous serosurveys. In June–July 2021, about 60% of children had evidence of antibodies against SARS-CoV-2. The findings of high seropositivity among children in our study are consistent with those of a recent study conducted in March–June 2021 among children from 5 sites in India [16]. Among the 700 individuals aged <18 years surveyed in this study, 55.7% were seropositive.

Earlier serosurveys indicated higher seroprevalence in urban slums and urban non-slum areas than in rural areas [5,6]. This gradient in seroprevalence seems to have faded as of June–July 2021, with comparable seroprevalence in rural and urban areas. The increased seroprevalence in rural areas observed in our survey indicates that infection in the second wave was widespread in rural areas.

Phase 2 vaccination trials reported that 98.4% (95% CI 95.3% to 99.7%) of individuals vaccinated with BBV152 (Covaxin) [17] and 100% (95% CI 97.4% to 100.0%) of individuals vaccinated with Covishield [18] had seroconverted by 56 days after the second dose. An observational multi-centre study among HCWs in India reported a seropositivity of 98.1% among Covishield recipients and 80% among Covaxin recipients [19]. In another study of antibody responses in a cohort of 45,965 adults from the general population in the United Kingdom who received either the ChAdOx1 nCoV-19 (Covishield) or BNT162b2 (Pfizer–

BioNTech) vaccine, authors estimated that about 6% of the participants were 'low responders' [20]. In our serosurvey, about 10%–14% of vaccinated individuals were found to be seronegative even after receiving 2 doses of COVID-19 vaccine. The proportion of seronegative individuals did not change when we considered only those individuals with the optimal interval between the second dose and sample collection of 7 days or more. Because of the cross-sectional nature of the study, we are not able to comment on whether this seronegativity among fully vaccinated individuals was due to lower antibody response or a decline in antibodies. Moreover, the possibility of misclassification of vaccination status cannot be ruled out, as the information about vaccination was based on recall.

The estimated number of infections per reported case did not change between December 2020–January 2021 (27.1–26.7) and June–July 2021 (21.7–27.8). This reflects sustained testing (around 210 million tests conducted between January-June 2021) of both symptomatic and asymptomatic eligible individuals [6].

Seroprevalence studies can help predict the future course of the pandemic [3]. Prior to the second wave of COVID-19 in India, about 75% of the population was seronegative [6]. The serosurvey findings indicate that about one-third of the general population in India did not have detectable antibodies against SARS-CoV-2 by June–July 2021. It is therefore likely that more COVID-19 cases will occur in coming months, especially in areas where the proportion of people without detectable antibodies is higher. The available evidence indicates that immunity acquired through natural infection can last up to 1 year [21]. IgG antibodies against S1-RBD show a high correlation with virus neutralisation titres, indicating the neutralising nature of the antibodies [22,23]. Studies also indicate that re-infections among previously infected individuals are less frequent [24]. It is thus reasonable to expect that a future surge of cases in India would be smaller than the second wave. However, the immunity acquired through natural infection as well as vaccination is expected to wane over time. Although S1-RBD antibodies are considered to have a neutralising effect, the protective titre among the seropositive individuals is not known. Further, COVID-19 cases could rapidly increase after the emergence of escape variants [25]. It is therefore necessary to continue monitoring the emergence of variants of concern.

Our study has certain limitations. First, our serosurvey was designed to estimate seroprevalence at the national level and might not have captured variation in seroprevalence within states and districts. Second, approximately 19% of eligible individuals were not included in the survey because they were not available in the household at the time of survey or they refused to participate. The age and gender distribution of the individuals who participated and who did not participate in the survey was different (S7 Table). This, however, would not have affected our seroprevalence estimates, as the age structure of the surveyed population was comparable to that of India's population. Third, IgG antibodies against SARS-CoV-2 wane over time [14,26]. In our study, seropositivity to S1-RBD and nucleocapsid protein among unvaccinated individuals with laboratory-confirmed COVID-19 was 82.4% and 63.4%, respectively. Hence, the observed seroprevalence might be an underestimate of the actual seroprevalence in the population.

In conclusion, our serosurvey findings indicate that nearly two-thirds of individuals aged ≥6 years from the general population and 85% of HCWs had antibodies against SARS-CoV-2 as of June–July 2021 in India. As one-third of the population was still seronegative, it is necessary to accelerate the coverage of COVID-19 vaccination among adults. COVID-19 cases in India have been declining since May 2021. However, continued surveillance for COVID-19 cases is necessary to detect an upsurge of COVID-19 cases early. Ongoing genomic surveillance for SARS-CoV-2 also needs to be strengthened to provide information about the emergence of newer variants, including their ability to circumvent immunity conferred by natural infection as well as vaccination. Finally, the high seroprevalence observed in the general

population should not be a reason for complacency. We believe that there is a need for continued adherence to non-pharmaceutical interventions, such as avoiding gatherings, ensuring social distancing, and using face masks in public places, to further reduce the transmission of SARS-CoV-2 in India.

## Supporting information

**S1 STROBE Checklist. STROBE checklist.**
(DOCX)

**S1 Appendix.**
(DOCX)

**S1 Protocol. Study protocol.**
(PDF)

**S1 Table. Seroprevalence of IgG antibodies against nucleocapsid protein by selected characteristics, June–July 2021.**
(DOCX)

**S2 Table. Cluster-adjusted proportion of individuals with SARS-CoV-2 IgG antibodies by state, June–July 2021.**
(DOCX)

**S3 Table. Unweighted proportion of individuals with SARS-CoV-2 IgG antibodies by district, June–July 2021.**
(DOCX)

**S4 Table. Unweighted proportion of individuals with SARS-CoV-2 IgG antibodies among unvaccinated individuals aged ≥10 years by district during the third (December 2020–January 2021) and fourth (June–July 2021) national serosurveys.**
(DOCX)

**S5 Table. Estimated number of SARS-CoV-2 infections among individuals aged 6 years and above and infection-to-case ratio.**
(DOCX)

**S6 Table. Comparison of seroprevalence of IgG antibodies against nucleocapsid protein and/or S1-RBD of SARS-CoV-2 by demographic characteristics between the third (December 2020–January 2021) and fourth (June–July 2021) national serosurveys.**
(DOCX)

**S7 Table. Characteristics of individuals who participated and who did not participate in the serosurvey, June–July 2021.**
(DOCX)

**S1 Text. Data analysis.**
(DOCX)

## Acknowledgments

ICMR serosurveillance group: Alka Turuk (Indian Council of Medical Research, New Delhi), Vikas Sabaharwal (National TB Elimination Programme, Uttarakhand), Nivethitha N. Krishnan (ICMR–National Institute for Research in Tuberculosis, Chennai, Tamil Nadu), Aby Robinson (ICMR–National Institute for Research in Tuberculosis, Chennai, Tamil Nadu), Nivetha

Srinivasan (ICMR–National Institute for Research in Tuberculosis, Chennai, Tamil Nadu), P. K. Anand (ICMR–National Institute for Implementation Research on Non-Communicable Diseases, Jodhpur, Rajasthan), Rushikesh Andhalkar (ICMR–National Institute for Research in Tuberculosis, Chennai, Tamil Nadu), Nimmathota Arlappa (ICMR–National Institute of Nutrition, Hyderabad, Telangana), Khalid Bashir (Government Medical College, Srinagar, Srinagar, Jammu and Kashmir), Pravin Bharti (ICMR–National Institute of Research in Tribal Health, Jabalpur, Madhya Pradesh), Debdutta Bhattacharya (ICMR–Regional Medical Research Centre, Bhubaneswar, Odisha), Sthita Pragnya Behera (ICMR–Regional Medical Research Centre, Gorakhpur, Uttar Pradesh), Ashrafjit S. Chahal (State TB Training and Demonstration Centre, Patiala, Punjab), Anshuman Chaudhury (ICMR–National Institute for Research in Tuberculosis, Chennai, Tamil Nadu), Alok Kumar Deb (ICMR–National Institute of Cholera and Enteric Diseases, Kolkata, West Bengal), Hirawati Deval (ICMR–Regional Medical Research Centre, Gorakhpur, Gorakhpur, Uttar Pradesh), Sarang Dhatrak (ICMR–National Institute of Occupational Health, Ahmedabad, Gujarat), Vikas Dhikav (ICMR–National Institute for Implementation Research on Non-Communicable Diseases, Jodhpur, Rajasthan), Rakesh Dayal (State TB Training and Demonstration Centre, Ranchi, Jharkhand), Prathiksha Giridharan (ICMR–National Institute for Research in Tuberculosis, Chennai, Tamil Nadu), Inaamul Haq (Government Medical College, Srinagar, Srinagar, Jammu and Kashmir), Babu Jagjeevan (ICMR–National Institute of Nutrition, Hyderabad, Telangana), Arshad Kalliath (State TB Training and Demonstration Centre, Thiruvananthapuram, Kerala), Srikanta Kanungo (ICMR–Regional Medical Research Centre, Bhubaneswar, Odisha), T. Karunakaran (ICMR–National Institute of Epidemiology, Chennai, Tamil Nadu), Jaya Singh Kshatri (ICMR–Regional Medical Research Centre, Bhubaneswar, Odisha), Niraj Kumar (ICMR–Regional Medical Research Centre, Gorakhpur, Uttar Pradesh), Vijay Kumar (ICMR–National Institute of Cancer Prevention and Research, Noida, Uttar Pradesh), V. G. Vinoth Kumar (ICMR–National Institute for Research in Tuberculosis, Chennai, Tamil Nadu), G. G. J. Naga Lakshmi (State TB Office, Hyderabad, Andhra Pradesh), Ganesh Mehta (ICMR–National JALMA Institute for Leprosy & Other Mycobacterial Diseases, Agra, Uttar Pradesh), Anindya Mitra (State TB Training and Demonstration Centre, Ranchi, Jharkhand), K. Nagbhushanam (ICMR–National Institute for Research in Tuberculosis, Chennai, Tamil Nadu), A. R. Nirmala (Lady Willingdon State TB Centre, Bengaluru, Karnataka), Subrata Kumar Palo (ICMR–Regional Medical Research Centre, Bhubaneswar, Odisha), Jaya Rameshwari Prabhu (State TB Office, Hyderabad, Andhra Pradesh), Ganta Venkata Prasad (State TB Office, Hyderabad, Andhra Pradesh), Uday Kumar Pucha (ICMR–National Institute of Nutrition, Hyderabad, Telangana), Mariya Amin Qurieshi (Government Medical College, Srinagar, Srinagar, Jammu and Kashmir), Seema Sahay (ICMR–National AIDS Research Institute, Pune, Maharashtra), Ramesh Kumar Sangwan (ICMR–National Institute for Implementation Research on Non-Communicable Diseases, Jodhpur, Rajasthan), Rochak Saxena (State TB Training and Demonstration Centre, Raipur, Chhattisgarh), Krithikaa Sekar (ICMR–National Institute for Research in Tuberculosis, Chennai, Tamil Nadu), Vijay Kumar Shukla (State TB Training and Demonstration Centre, Raipur, Chhattisgarh), Hari Bhan Singh (ICMR–National JALMA Institute for Leprosy & Other Mycobacterial Diseases, Agra, Uttar Pradesh), Prashant Kumar Singh (ICMR–National Institute of Cancer Prevention and Research, Noida, Uttar Pradesh), Pushpendra Singh (ICMR–National Institute of Research in Tribal Health, Jabalpur, Madhya Pradesh), Rajeev Singh (ICMR–Regional Medical Research Centre, Gorakhpur, Uttar Pradesh), Mahendra Thakor (ICMR–National Institute for Implementation Research on Non-Communicable Diseases, Jodhpur, Rajasthan), Ankit Viramgami (ICMR–National Institute of Occupational Health, Ahmedabad, Gujarat), Pradeep A. Menon (ICMR–National Institute for Research in Tuberculosis, Chennai, Tamil Nadu), Rajiv Yadav (ICMR–

National Institute of Research in Tribal Health, Jabalpur, Madhya Pradesh), Surabhi Yadav (ICMR–National Institute for Research in Tuberculosis, Chennai, Tamil Nadu), Manjula Singh (Indian Council of Medical Research, New Delhi), Amit Chakrabarti (ICMR–National Institute of Occupational Health, Ahmedabad, Gujarat), Aparup Das (ICMR–National Institute of Research in Tribal Health, Jabalpur, Madhya Pradesh), Shanta Dutta (ICMR–National Institute of Cholera and Enteric Diseases, Kolkata, West Bengal), Rajni Kant (ICMR–Regional Medical Research Centre, Gorakhpur, Uttar Pradesh), A. M. Khan (Indian Council of Medical Research, New Delhi), Kanwar Narain (ICMR–Regional Medical Research Centre, N. E. Region, Dibrugarh, Assam), Somashekar Narasimhaiah (National Tuberculosis Institute, Bangalore, Karnataka), Chandrasekaran Padmapriyadarshini (ICMR–National Institute for Research in Tuberculosis, Chennai, Tamil Nadu), Krishna Pandey (ICMR–Rajendra Memorial Research Institute of Medical Sciences, Patna, Bihar), Sanghamitra Pati (ICMR–Regional Medical Research Centre, Bhubaneswar, Odisha), Hemalatha Rajkumar (ICMR–National Institute of Nutrition, Hyderabad, Telangana), Arun Kumar Sharma (ICMR–National Institute for Implementation Research on Non-Communicable Diseases, Jodhpur, Rajasthan), Y. K. Sharma (State TB Training and Demonstration Centre, Raipur, Chhattisgarh), Shalini Singh (ICMR–National Institute of Cancer Prevention and Research, Noida, Uttar Pradesh), Samiran Panda (Indian Council of Medical Research, New Delhi), D. C. S. Reddy (Banaras Hindu University, Varanasi, Uttar Pradesh).

The authors gratefully acknowledge the technical inputs provided by the Epidemiology and Surveillance Working Group of the ICMR COVID-19 National Task Force (S1 Appendix). The authors also thank state-level survey teams for their field work and the WHO Country Office for India, state and district health officials, and primary healthcare staff for coordination of field operations. We thank the laboratory and data management teams at ICMR–National Institute of Epidemiology, Chennai (S1 Appendix).

## Author Contributions

**Conceptualization:** Manoj V. Murhekar, Tarun Bhatnagar, Jeromie Wesley Vivian Thangaraj, Muthusamy Santhosh Kumar, Kiran Rade, Balram Bhargava.

**Data curation:** V. Saravanakumar, R. Sabarinathan.

**Formal analysis:** Jeromie Wesley Vivian Thangaraj, V. Saravanakumar.

**Funding acquisition:** Manoj V. Murhekar.

**Investigation:** Sriram Selvaraju, C. P. Girish Kumar, Smita Asthana, Rakesh Balachandar, Sampada Dipak Bangar, Avi Kumar Bansal, Jyothi Bhat, Debjit Chakraborty, Vishal Chopra, Dasarathi Das, Kangjam Rekha Devi, Gaurav Raj Dwivedi, Agam Jain, S. Muhammad Salim Khan, M. Sunil Kumar, Avula Laxmaiah, Major Madhukar, Amarendra Mahapatra, Chethana Rangaraju, Jyotirmayee Turuk, Suresh Yadav.

**Methodology:** Manoj V. Murhekar, Tarun Bhatnagar, Jeromie Wesley Vivian Thangaraj, Muthusamy Santhosh Kumar.

**Project administration:** Manoj V. Murhekar, Jeromie Wesley Vivian Thangaraj, Muthusamy Santhosh Kumar, Sriram Selvaraju, Kiran Rade, C. P. Girish Kumar, Smita Asthana, Rakesh Balachandar, Sampada Dipak Bangar, Avi Kumar Bansal, Jyothi Bhat, Debjit Chakraborty, Vishal Chopra, Dasarathi Das, Kangjam Rekha Devi, Gaurav Raj Dwivedi, Agam Jain, S. Muhammad Salim Khan, M. Sunil Kumar, Avula Laxmaiah, Major Madhukar, Amarendra Mahapatra, Talluri Ramesh, Chethana Rangaraju, Jyotirmayee Turuk, Suresh Yadav, Balram Bhargava.

**Resources:** Sriram Selvaraju, Kiran Rade, C. P. Girish Kumar, Smita Asthana, Rakesh Balachandar, Sampada Dipak Bangar, Avi Kumar Bansal, Jyothi Bhat, Debjit Chakraborty, Vishal Chopra, Dasarathi Das, Kangjam Rekha Devi, Gaurav Raj Dwivedi, Agam Jain, S. Muhammad Salim Khan, M. Sunil Kumar, Avula Laxmaiah, Major Madhukar, Amarendra Mahapatra, Talluri Ramesh, Chethana Rangaraju, Jyotirmayee Turuk, Suresh Yadav, Balram Bhargava.

**Software:** R. Sabarinathan.

**Supervision:** Jeromie Wesley Vivian Thangaraj, Muthusamy Santhosh Kumar, Sriram Selvaraju, Kiran Rade, C. P. Girish Kumar, R. Sabarinathan, Smita Asthana, Rakesh Balachandar, Avi Kumar Bansal, Jyothi Bhat, Debjit Chakraborty, Vishal Chopra, Dasarathi Das, Kangjam Rekha Devi, Gaurav Raj Dwivedi, Agam Jain, S. Muhammad Salim Khan, M. Sunil Kumar, Avula Laxmaiah, Major Madhukar, Amarendra Mahapatra, Talluri Ramesh, Chethana Rangaraju, Jyotirmayee Turuk, Suresh Yadav.

**Validation:** R. Sabarinathan.

**Visualization:** V. Saravanakumar.

**Writing – original draft:** Manoj V. Murhekar, Jeromie Wesley Vivian Thangaraj.

**Writing – review & editing:** Manoj V. Murhekar, Tarun Bhatnagar, Jeromie Wesley Vivian Thangaraj, V. Saravanakumar, Muthusamy Santhosh Kumar, Sriram Selvaraju, Kiran Rade, C. P. Girish Kumar, R. Sabarinathan, Smita Asthana, Rakesh Balachandar, Sampada Dipak Bangar, Avi Kumar Bansal, Jyothi Bhat, Debjit Chakraborty, Vishal Chopra, Dasarathi Das, Kangjam Rekha Devi, Gaurav Raj Dwivedi, Agam Jain, S. Muhammad Salim Khan, M. Sunil Kumar, Avula Laxmaiah, Major Madhukar, Amarendra Mahapatra, Talluri Ramesh, Chethana Rangaraju, Jyotirmayee Turuk, Suresh Yadav, Balram Bhargava.

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
