## [Editor Report · Decision Letter 0]

11 Aug 2021

Dear Dr Murhekar, 

Thank you for submitting your manuscript entitled "Prevalence of IgG antibodies against SARS-CoV-2 among the general population and healthcare workers in India, June–July 2021" for consideration by PLOS Medicine.

Your manuscript has now been evaluated by the PLOS Medicine editorial staff and I am writing to let you know that we would like to send your submission out for external assessment.

However, we first need you to complete your submission by providing the metadata that is required for full assessment. To this end, please login to Editorial Manager where you will find the paper in the 'Submissions Needing Revisions' folder on your homepage. Please click 'Revise Submission' from the Action Links and complete all additional questions in the submission questionnaire.

Please re-submit your manuscript within two working days, i.e. by Aug 13 2021 11:59PM.

Once your full submission is complete, your paper will undergo a series of checks in preparation for assessment. 

Kind regards,

Richard Turner, PhD

rturner@plos.org

---

## [Decision Letter · Decision Letter 1]

8 Oct 2021

Dear Dr. Murhekar,

Thank you very much for submitting your manuscript "Prevalence of IgG antibodies against SARS-CoV-2 among the general population and healthcare workers in India, June–July 2021" (PMEDICINE-D-21-03454R1) for consideration at PLOS Medicine. 

Your paper was discussed with an academic editor with relevant expertise and sent to independent reviewers, including a statistical reviewer. The reviews are appended at the bottom of this email and any accompanying reviewer attachments can be seen via the link below:

[LINK]

In light of these reviews, we will not be able to accept the manuscript for publication in the journal in its current form, but we would like to invite you to submit a revised version that addresses the reviewers' and editors' comments fully. You will appreciate that we cannot make a decision about publication until we have seen the revised manuscript and your response, and we expect to seek re-review by one or more of the reviewers. 

We hope to receive your revised manuscript by Oct 28 2021 11:59PM. Please email us (plosmedicine@plos.org) if you have any questions or concerns.

Please let me know if you have any questions, and we look forward to receiving your revised manuscript. 

Sincerely,

Richard Turner PhD

Senior editor, PLOS Medicine

rturner@plos.org

We ask you to limit the number of authors to fewer than 30, with the additional contributors forming an author group.

Noting PLOS' data policy (https://journals.plos.org/plosmedicine/s/data-availability), please confirm that de-identified study data will be made available, and supply a non-author contact for those interested in inquiring about data access. 

Please adapt the title to include a study descriptor following a colon, e.g., "... June-July, 2021: A population-based cohort study".

In the abstract and throughout the paper, please quote p values alongside 95% CI, where available.

Please trim the "Conclusions" subsection of your abstract to around half the current length. 

After the abstract, please add a new and accessible "Author summary" in non-identical prose. You may find it helpful to consult one or two recent research papers published in PLOS Medicine to get a sense of the preferred style. 

In the methods section, please state whether the study had a protocol or prespecified analysis plan, and if so attach the document as a supplementary file, referred to in the text. 

Please highlight analyses that were not prespecified. 

Throughout the text, please adapt reference call-outs to the following style: "... cases reported globally [1].".

Please remove the information on competing interests and data sharing from the end of the main text. In the event of publication this will appear in the article metadata, via entries in the submission form. 

Please move the list of contributors at the end of the ms to a supplementary file.

In the reference list, please convert italics and boldface text into plain text. 

Where appropriate, 6 author names should be listed, followed by "et al.".

Please add a completed checklist for the most appropriate reporting guideline, e.g., STROBE, as an attachment, labelled "S1_STROBE_Checklist" or similar and referred to as such in your Methods section. 

In the checklist, please refer to individual items by section, e.g., "Methods", and paragraph numbers, not by line or page numbers as these generally change in the event of publication. 

Comments from the reviewers:

*** Reviewer #1: 

This work is one of the extensive work done during the time of covid-19 and particularly from countries where drastic infection has occurred.

Very limited layout or format comments.

1- I like the title of "Seroprevalance" in this case than "prevalence", as testing was used to detect the antibodies rather than other type of testing. 

2- abstract: the paragraph related to the limitations, could be reflected clearly under subheadings in the discussion, as it was done. no need to be in the abstract.

3- abstract, in the conclusion: the second line, "by June 2020". I think it should be stated as "June 2021".

4- abstract: The conclusion is so long and should be reduced to the main points. No matter to be little wider in the text. 

5- Result, page 7: The paragraph started with "the weighted prevalence of IgG antibodies ...........(table 2)", probably should be stated when the description of table 2 is done. 

6- References, page 11: Format and layout of the references is required according to the Journal instructions. 

7- I recommend to use the table related to the Health care workers to be among the main tables in the text and not among the supplementary. 

*** Reviewer #2: 

Abstract: in conclusion there is typo error ".... by June 2020" 

Methods: sample size calculation with design effect should be provided for the general population and also for the HCWs. 

*** Reviewer #3:

[supportive report received] 

*** Reviewer #4: 

Important study showing the power of infection and capacity of transmission this virus in a different population (non-HCW and HCW) living in different regions (rural, non-slum, and -slum areas) of a big country. The authors dedicate strong space to describe the methods used, however, I would like to suggest improving the description of the regression model, including why the authors did not use Poisson Regression model in this cross-sectional study?! On the other hand, I understood the use of weights design mainly because of the probable difference among the size population of the district studied, however, the authors should have provided in the tables (#3, and #4 and #5) the target-population, this is important to share with the readers the probable high variability among the clusters represented of the districts studied.

*** Reviewer #5: 

[See attachment]

Michael Dewey

***

[LINK]

---

## [Decision Letter · Decision Letter 2]

17 Nov 2021

Dear Dr. Murhekar,

Thank you very much for re-submitting your manuscript "Seroprevalence of IgG antibodies against SARS-CoV-2 among the general population and healthcare workers in India, June–July 2021: A population-based cross-sectional study" (PMEDICINE-D-21-03454R2) for consideration at PLOS Medicine.

I have discussed the paper with our academic editor and it was also seen again by two reviewers. I am pleased to tell you that, provided the remaining editorial and production issues are fully dealt with, we expect to be able to accept the paper for publication in the journal.

[LINK]

Please let me know if you have any questions, and we look forward to receiving the revised manuscript.   

Sincerely,

Richard Turner, PhD

rturner@plos.org

Requests from Editors:

Please adapt the data sharing statement (submission form) to read "... Given the nature of the data, potential users will be asked to sign a data-sharing agreement".

As mentioned previously, please include no more than 30 authors, with the remaining individuals participating as a named author group, with group members listed in the Acknowledgements. 

We ask you to add an additional sentence, say, to the abstract to quote additional information on the cohort studied (quoting the number of participants and some information on age, sex and residence). 

Please make that "test-adjusted" in the abstract, and throughout the text where the phrase is used as an adjective. 

Please revisit the "Author summary" section, aiming for no more than 3-4 points in each of the three subsections. For example, the final point of the first subsection ("We conducted the fourth ...") should either be deleted or moved to the second subsection. 

At the end of the Introduction (main text), please make that "We conducted the fourth round of a national ...".

Under editorial query "8" you mention non-prespecified sensitivity analyses. Please add a few words to the methods and/or results sections of the paper, if not already present, to highlight these non-prespecified analyses. 

In the second paragraph of the Discussion section (main text) there is a superscript "2". Please reformat this if it is a reference call-out. 

Noting reviewer 5's comment, we ask you to add "we believe" or similar when you are making recommendations not directly based on the findings of your study. 

Please spell out the institutional author name for reference 10.

Noting references 15 & 16, please add "[preprint]" to all preprints cited, unless you are able to substitute the corresponding peer-reviewed papers. 

Comments from Reviewers:

*** Reviewer #4: 

I believe is very important to discuss in the limitation section the option related to use in the analysis a mixed-models GLM with overdispersion of the data, mainly when it's applied in cross sectional study.

*** Reviewer #5: 

The authors have addressed my points. We still disagree about the wisdom of including recommendations (genomic scanning and mask wearing) in the conclusion which are not supported by the authors' research however sensible they may be. I think this is perhaps more of an editorial policy issue so I leave it to the team to decide.

Michael Dewey

***

[LINK]

---

## [Editor Report · Decision Letter 3]

29 Nov 2021

Dear Dr Murhekar, 

On behalf of my colleagues and the Academic Editor, Dr Suthar, I am pleased to inform you that we have agreed to publish your manuscript "Seroprevalence of IgG antibodies against SARS-CoV-2 among the general population and healthcare workers in India, June–July 2021: A population-based cross-sectional study" (PMEDICINE-D-21-03454R3) in PLOS Medicine.

Prior to final acceptance, please: 

- Regarding the following sentence in the author summary: "Considering high seroprevalence of SARS-CoV-2, the future surge of cases in India is expected to be lower than the second wave.", please amend the sentence to "The substantial seroprevalence of anti-SARS-CoV-2 antibodies in the Indian population should provide a measure of protection against future waves of COVID-19 in the country.", or similar; 

- Reverse the order of the two points in the "What do these findings mean?" in the author summary; and 

- Move "[preprint]" from the Discussion section to the appropriate entry in the reference list (reference 16). 

PRESS

Sincerely, 

Richard Turner, PhD 

rturner@plos.org